# Advances in HIV Gene Therapy

**DOI:** 10.3390/ijms25052771

**Published:** 2024-02-28

**Authors:** Rose Kitawi, Scott Ledger, Anthony D. Kelleher, Chantelle L. Ahlenstiel

**Affiliations:** 1Kirby Institute, University of New South Wales, Kensington, NSW 2052, Australia; rkitawi@kirby.unsw.edu.au (R.K.); akelleher@kirby.unsw.edu.au (A.D.K.); 2St. Vincent’s Hospital, Darlinghurst, NSW 2010, Australia; 3UNSW RNA Institute, University of New South Wales, Kensington, NSW 2052, Australia

**Keywords:** ex vivo, gene therapy, stem cells, vector, HIV

## Abstract

Early gene therapy studies held great promise for the cure of heritable diseases, but the occurrence of various genotoxic events led to a pause in clinical trials and a more guarded approach to progress. Recent advances in genetic engineering technologies have reignited interest, leading to the approval of the first gene therapy product targeting genetic mutations in 2017. Gene therapy (GT) can be delivered either in vivo or ex vivo. An ex vivo approach to gene therapy is advantageous, as it allows for the characterization of the gene-modified cells and the selection of desired properties before patient administration. Autologous cells can also be used during this process which eliminates the possibility of immune rejection. This review highlights the various stages of ex vivo gene therapy, current research developments that have increased the efficiency and safety of this process, and a comprehensive summary of Human Immunodeficiency Virus (HIV) gene therapy studies, the majority of which have employed the ex vivo approach.

## 1. Introduction

Successful sequencing of the human genome [1] has underpinned an increased understanding of heritable diseases and pathogenic processes that cause chronic and transient disease conditions. More than 5000 gene therapy products are currently under trial for various hereditary, chronic and infectious diseases (https://clinicaltrials.gov/, accessed on 30 October 2023). In the development of a gene therapy candidate, it is important to consider an efficient method of delivery that will allow accurate processing of the gene of interest, with sufficient expression at the specific site of action and to the specific site of action. It is also important to sustain the therapeutic effect for the intended duration and to subsequently degrade the vehicle of delivery without causing any untoward effects. Due to the potential danger of causing serious adverse effects like oncogenesis and even germline transmission of the gene modification, the process of developing gene therapy products (GTP) is very rigorous and takes longer than other therapeutic agents. Appendix A gives a summary of gene and cell products that have been approved so far by different regulatory authorities [2,3,4,5,6,7,8,9,10,11,12,13,14,15,16,17,18,19,20,21,22,23,24,25,26,27,28,29,30,31,32,33,34]. A review has been published recently on the development of some of these products [35]. There are two approaches to delivering the gene of interest: in vivo and ex vivo (Figure 1). The in vivo approach involves delivery of the genetic material directly to the body of the patient, either as the naked gene or encapsulated in a particle. On the other hand, the ex vivo approach involves the isolation of the cells of interest from the patient or a normal donor, genetically modifying them, expanding them in some instances, and then administering them to the patient. The patient therefore has no direct contact with the transfer vector [36]. Selection, that is, ensuring that the modified cells are preferentially engrafted in the patient as opposed to the naturally occurring deficient cells, can be undertaken before or after administering the gene of interest [36]. A number of reviews have recently been published that highlight the applications of gene therapy in various fields: using hematopoietic stem and progenitor cells (HSPCs) for hematopoietic disorders [37,38], genetic skin diseases [39], neurological disorders or genetic diseases of the central nervous system [40,41] and orthopaedic diseases [42], among others.

## 2. Cure Approaches of HIV Gene Therapy

Obtaining a cure for HIV has been challenging due to the latent viral reservoir, which is resistant to current therapy. It is transcriptionally inactive, but a significant portion is replication competent. Antiretroviral therapy (ART) may be relatively poorly absorbed into tissues that harbour the latent virus, like the gut-associated lymphoid tissue (GALT) and the central nervous system (CNS), which may aid viral persistence in these sites. Viral rebound is observed within weeks to months of cessation of ART. Another challenge to cure is the effect of HIV on the immune system: there is a poor humoral response to HIV with inadequate production of neutralizing antibodies, which only occurs after several months of infection. Besides, there is also a relative failure to reconstitute HIV-specific CD4^+^ T cells, which in turn leads to the reduced function of CD8^+^ cytotoxic T cells [43].

It is therefore reasonable to postulate that an effective cure will aim at inactivating or removing the latent virus and reconstituting the immune system. Gene-modified cells should also be resistant to ongoing infection, and the therapy itself should not cause severe adverse events. To reconstitute or enhance the function of the immune system, there have been several developments in the genetic engineering of B and T cells, including chimeric antigen receptor T (CAR-T) cell therapy. These developments have recently been reviewed [43,44]. Figure 2 provides a summary of HIV cure strategies.

A multi-stakeholder consultation by the International AIDS Society agreed on minimum criteria for a target product for HIV cure. Clinical efficacy was defined as follows: the maintenance of the viral load below the transmission threshold (<200 HIV RNA copies/mL), efficacy in 20% or more of the population under study, an average relapse rate of less than 10% a year, a remission duration of greater than 2 years and minimal chances of serious adverse effects, if any, i.e., less than 1% of the population under study. A possible cure also is likely to require the combination of different approaches [45].

There are four strategies that have been employed to achieve a possible cure for HIV. The first approach, stem cell transplantation, has resulted in the cure of six patients so far: the Berlin patient, the London patient, the New York patient, the ‘City of Hope patient’, the Dusseldorf patient and the Geneva patient. These patients underwent hematopoietic stem cell transplantation to treat malignancies. In all cases, except the Geneva patient, the patients were transplanted with human leukocyte antigens (HLA)-matched stem cells from a donor homozygous for a 32 base pair deletion in the CCR5 allele (CCR5Δ32) [46,47,48]. This approach cannot be scaled to a larger population due to the aggressive nature of the treatment, cost, morbidity and mortality, as well as a lack of matched unrelated donors with CCR5Δ32, and is therefore not feasible as a generalizable approach.

The second approach aims to remove all viable latent virus from cell reservoirs, achieving an undetectable plasma load without needing ART [49], hence providing a sterilizing cure. This strategy is called the ‘Shock and Kill’ approach and has been studied extensively in early phase trial [50,51,52]. Although current forms of this cure approach do not employ gene therapy, it has been included in this review, in order to cover all current HIV cure approaches. ‘Shock and Kill’ aims to activate the virus in the latent reservoir using latency reversing agents, which activate virus transcription, prevent infection of new cells by use of concurrent ART and then eradicate the infected cells by inducing apoptosis, CD8^+^ mediated lysis or through the humoral immune response [53]. Chimeric Antigen Receptor T-cells (CAR-T) cells could be also employed to enhance the killing of infected cells [54].

The third strategy aims to eradicate the integrated virus through gene editing, using nuclease-based tools or engineered recombinase enzymes [55]. Nuclease-based tools include CRISPR (clustered regularly interspaced short palindromic repeats) [56], TALEN (transcription-activator-like effector nucleases) [57] and ZFN (zinc-finger nucleases) [58]. It is also possible to target CCR5 using these nuclease-based tools to render CD4^+^ cells resistant to infection [59], as well as silencing CCR5 using short interfering or short hairpin RNA (siRNA or shRNA [60]). Gene editing can also occur through engineered recombinase enzymes, specifically, the HIV-specific long terminal repeat (LTR) recombinase enzymes- tailored site specific (TRE) recombinase. This is an engineered form of Cre recombinase that targets a 34 bp region within the 5′LTR (known as the loxLTR), resulting in the removal of integrated proviral DNA in infected cells expressing this enzyme [55,61].

The fourth strategy is the ‘Block and Lock’ approach, which aims to achieve a functional cure where the latent virus is maintained in its inactive state without eradicating it, with the plasma viral load remaining below detectable levels [49]. This is achieved by permanently silencing or editing the latent reservoir using latency promoting agents that prevent or ‘block’ virus transcription and ‘lock’ the virus in a latent state, through repressive epigenetic modifications of the viral promoter [53,62,63,64,65,66].

The majority of HIV gene therapy studies use the ex vivo approach. This review will focus on the various stages of ex vivo gene therapy and provide a summary of the current HIV clinical trials that have employed ex vivo or in vivo gene therapy approaches.

## 3. Ex Vivo Gene Therapy

An important factor to consider in ex vivo gene therapy studies is the source of cells that will be modified to correct a particular disease condition or genetic disorder. These cells could be autologous, being obtained from the patient themselves, or allogenic being obtained from a matched donor. Autologous sources are advantageous, as there is no danger of graft-versus-host disease; however, this may not be suitable for older patients or severe disease conditions (e.g., HIV, as the cells may be a source of new infection). Allogenic stem cell sources have attracted greater interest recently due to their capacity to self-renew and differentiate to several lineages, although there are limitations in their use, such as rejection, graft-versus-host disease and prolonged immunodeficiency [67]. In pre-clinical gene therapy studies, stem cells can be obtained from birth tissue, e.g., umbilical cord blood and placenta. Stem cells can also be obtained from embryonic tissue, although this raises ethical concerns, or from induced pluripotent stem cells. In clinical studies, cells can be obtained either from the individual patient, depending on their disease condition, or from a donor whose cells are immunologically compatible with the patient’s tissues, or from other cell sources that have a low chance of evoking an immune response, such as cells of placental origin due to the lack of or low expression of HLA Class I and II antigens [68]. The process of ex vivo gene therapy involves obtaining these cells of interest, modifying them, and administering them to the patient. To ensure the efficient and preferential engraftment of the modified cells, various conditioning and selection methods are used.

### 3.1. Sources of Hematopoietic Stem Cells

Hematopoietic stem cells are useful in gene therapy for monogenic disorders and diseases of the immune system. Hematopoietic stem cells (HSCs) and hematopoietic progenitor cells (HPCs) are responsible for the production of adult blood cells and are characterised by specific markers (Figure 3). The CD34 antigen, a cell surface adhesion transmembrane glycoprotein molecule, has long been used as a marker for HSCs, as well as having other functions like enhancing cell proliferation, blocking differentiation, improving the migration of HSCs and HPCs and promoting lymphocyte adhesion to the vascular endothelium in lymphoid tissues [69]. These cells have the ability to differentiate into various lineages of functional blood cells and are capable of self-renewal; i.e., they can generate daughter HSCs without differentiation (Figure 1) [70,71,72].

#### 3.1.1. Bone Marrow-Derived Hematopoietic Stem Cells

Common sources of HSCs for gene therapy application include bone marrow, umbilical cord, peripheral blood and more recently, placenta. In bone marrow, osteogenic cells influence the balance of haematopoiesis. Primary human osteoblasts are essential for the survival of bone marrow-derived CD34^+^ HSCs, due to osteoblasts constitutively expressing granulocyte cell stimulating factor (G-CSF) [73]. Similarly, HSCs regulate the secretion of IL-6, macrophage inflammatory protein-1α and other factors from osteoblasts, in a bid to create a conducive environment for haematopoiesis [74]. The stromal-derived factor-1 (SDF-1) and its receptor C-X-C chemokine receptor 4 (CXCR4) are important determinants for the establishment of bone marrow [75]. Tie2, a receptor tyrosine kinase, has also been found to be indispensable in maintaining HSCs in adult bone marrow [76]. The CD34 antigen has multiple phosphorylation sites, with two sites for protein kinase C and one site for tyrosine phosphorylation [77].

An article by Panch et al. [78] gives a summary on harvesting HSCs from bone marrow, peripheral blood and the umbilical cord. Briefly, harvesting HSCs from bone marrow is a procedure often performed under general anaesthesia, wherein approximately 20 mL per kg, and not more than 1.5 L, of marrow aspirate is collected from the posterior or anterior iliac crest. Before the procedure is performed, blood may be collected from the patient so that it can be re-infused to replace blood lost during aspiration.

#### 3.1.2. Peripheral Blood-Derived Hematopoietic Stem Cells

The procedure for harvesting HSCs from peripheral blood is performed over several days. Early mobilization regimens used cytotoxic agents, such as cyclophosphamide, idarubicin, etoposide, platinum and epirubicin; however, the use of these agents is now limited to patients who are undergoing transplantation to treat malignancy [79]. Currently, in transplant patients, granulocyte cell-stimulating factor (G-CSF), for the mobilization of CD34^+^ cells into the peripheral blood, is given at a dose of 5–10 mg/kg/day for 5 to 7 days, with the aim of achieving a target number of CD34^+^ cells harvested by leukapheresis of at least 2 × 10^6^ cells/kg body weight [80]. During the period of G-CSF administration, the concentration of HSCs in blood increases after 3 days, peaks around day 5 or 6 and then starts to fall after 7 days [81]. The dose of G-CSF is determined by the peripheral white blood cell count and the peripheral CD34^+^ count is a good predictor of the yield of HSCs [82]. Yield is also determined by age, sex, underlying condition and dose of G-CSF. G-CSF treatment is not without its challenges, with patients experiencing side effects including bone pain, headache, fatigue, myalgia and, in severe cases, myocardial infarction and cerebral ischemia in high-risk individuals [80]. To enhance the mobilizing effect of G-CSF, AMD3100 (Plerixafor), a reversible CXCR4 antagonist, may be used. CXCR4 is the receptor for stromal-derived factor 1 (SDF-1) which has numerous functions, including quiescence, which results in long term HSC maintenance [83]. Plerixafor is administered at a dose of 240 µg/kg, 4 to 6 h before apheresis [78] and has been found to improve the harvest of CD34^+^ cells, especially when used in combination with G-CSF. Other HSC mobilization agents are currently under development [84].

#### 3.1.3. Umbilical Cord-Derived Hematopoietic Stem Cells

Harvesting cells from the umbilical cord involves the venipuncture of the severed umbilical cord and draining the blood into a sterile bag with anticoagulant. Normally, approximately 80 to 160 mL of umbilical cord blood (UCB) is collected during the process. The yield of CD34^+^ cells ranges from 2 to 7 × 10^6^/mL UCB and can be affected by birth order (first born > second born > third born, etc.), birth weight [85] and even early clamping [86]. This relatively low yield limits the applications for which these can be used.

#### 3.1.4. Placenta-Derived Hematopoietic Stem Cells

The human placenta has gained popularity recently as a source of stem cells, because placental stem cells have low antigenicity, have no ethical restrictions and are pluripotent. The human placenta is the first organ to develop in the reproductive process and has two components, the foetal component (amnion and chorion) and the maternal component (the decidua). The development of the placenta, from the moment it appears, 6–7 days after fertilization, up to term, has been described in detail [87]. The placenta serves to provide nutrients to and remove waste from the developing foetus and has various secretory and immunomodulatory roles [88]. It can be divided into the following four regions: amniotic epithelial, amniotic mesenchymal, chorionic mesenchymal and chorionic trophoblastic [68]. The placenta has various types of stem cells that can differentiate into hematopoietic and mesenchymal tissues (adipogenic, chondrogenic, osteogenic, hepatic, pancreatic, myogenic, angiogenic and neurogenic), expressing various cell markers including OCT4, SOX2 and c-KIT, among others that are also found in embryonic stem cells [88,89]. Placental HSCs lack or have very low expressions of HLA class I antigens (HLA-A, HLA-B, HLA-C) and no HLA class 2 antigens (HLA-DP, HLA-DQ, HLA-DR) which make them useful for regenerative medicine and for autologous and allogenic transplantation. These cells have a reduced chance of immune rejection compared to bone marrow cells, are resistant to apoptosis, have enhanced cell proliferation and wound healing properties and inhibit pro-fibrotic factors like TGFβ [68]. Human placental HSCs are of foetal origin (not maternal) [90] and are CD34+- and CD45-dim. The perfusion of the human placenta with AMD3100 (Plerixafor), at a concentration of 300 ug/L, results in an increase (of more than six-fold) in the amount of HSCs harvested in the perfusate, and these have colony-forming properties and lack endothelial markers [90]. Mesenchymal stromal cells are an important component of the bone marrow hematopoietic environment and co-culturing them with HSCs has been found to enhance proliferation of HSCs, especially the primitive CD34^+^ and CD38^-^ [91,92]. Hematopoietic stem cell transplantation has been performed, using a mixture of cord blood and placenta-derived stem cells [93].

Various protocols have been developed to isolate the amniotic epithelial cells [94], mesenchymal stromal cells [95,96] and hematopoietic stem cells [97] from the placenta. One of the major hurdles that all protocols have to overcome is the production of a significant number of cells with a high level of purity of the desired cell population [98].

### 3.2. Isolation, Purification and Enrichment of Hematopoietic Stem Cells

To ensure the preferential engraftment of gene-corrected cells, it is important that the population of cells delivered is enriched for the cells with the construct and purified to avoid contamination with proteins or other molecules used during culturing. Several methods have been used for the enrichment of HSCs. The most common column separation technique currently is the magnetic-activated cell sorting (MACS^®^) microbead separation technology from Miltenyi Biotech, Cologne, Germany (https://www.miltenyibiotec.com/ accessed on 4 June 2023). In this closed system technology, microbeads (superparamagnetic iron oxide nanoparticles) are conjugated to CD34 antibodies for the magnetic labelling of cells expressing CD34+. The sterile cell suspension is loaded into a MACS^®^ (ferromagnetic) separation column which is placed in the magnetic field of a MACS^®^ Separator. In positive selection techniques, the labelled cells are retained in the column while the rest of the cells run through. The CD34+ cells can then be eluted from the column, once it is removed from the magnetic field generated by the Separator, for any downstream application. There are microbeads available also for the separation of the CD133+ cell population and CD34+ and CD38− cells. Baldwin and colleagues used immunomagnetic beads to enrich for CD34+ cells and thereafter fractionated them using fluorescence-activated cell sorting (FACS) to enrich for CD34+ and CD38− cells [99]; Radtke and colleagues, on the other hand, isolated CD34+, CD90+ and CD45RA− cells using FACS and found that this group of cells were phenotypically the most defined target for HSC gene therapy [100], as these drive short-term and long-term multilineage engraftment. They used a GMP (good manufacturing practice) grade flow-sorting protocol to isolate these cells. This adds considerably to the complexity, cost and expertise required. In other in vitro studies, Kays and colleagues used MACS to sort for cells which, in addition to being CD34+, also expressed CD105, a component of the transforming growth factor-β (TGF-β) receptor complex, and found that these cells were enriched for early HSCs that had a high engraftment and repopulating capacity even after lentiviral transduction [101]. Laje and colleagues enriched the HSCs destined for lentiviral transduction with the signalling lymphocyte activation molecule (SLAM) family of receptor molecules and found that these cells were effectively transduced and were stable compared to the non-transduced cells [102].

Studies have shown that culturing HSCs for more than 48 h reduces their long-term engraftment capacity [103,104] and it is recommended that cultures should not exceed 36 h, with 24 h being a safe timeframe [104]. A study by Chen and colleagues found that there is mitochondrial oxidative stress following ex vivo culturing with cytokines (SCF, TPO and Flt3L) that led to a loss of stemness [105] and that the adhesion GPCR G1–positive (ADGRG1^+^) population of CD34^+^/CD133^+^ can enrich for functional HSCs under oxidative stress during ex vivo culturing.

Efforts have also been made to reduce possible sources of contamination during culture. Recombinant human serum albumin (HSA), which is produced from yeast and is used as a component of GMP-grade tissue culture media, could be a source of contamination with other proteins that do not interact with albumin directly or even pigments and small molecules. Wilkinson and colleagues, using in vitro and in vivo mice models, recently found that polyvinyl alcohol (PVA) could be a substitute for HSA in the culturing of both mouse and human HSCs as it supports HSC survival and growth, whilst maintaining the phenotypic HSC characteristics. The PVA culture medium had lower concentrations of secreted factors and less senescence-associated gene expression [106,107,108,109], although the use of PVA for human gene therapy is yet to be assessed.

### 3.3. Modification of Hematopoietic Stem Cells with Genes of Interest

In considering an appropriate vector for gene delivery, one needs to examine the characteristics of the cell to be transduced and the characteristics of the vector; whether a transient or stable expression of the gene modification is required; and possible ‘off-target’ effects that can result upon delivery to tissue. These off-target effects can include host–immune reactions and immune reactions induced by the vector itself. The modification of target cells with the gene of interest can face various challenges. For example, adeno-associated virus vectors do not effectively transduce HSCs due to blocks to nuclear entry, uncoating and second-strand synthesis [110]. Pseudotyping lentiviral vectors with a VSV-G envelope leads to broad cell tropism although primary human resting T cells, and CD34+ cells are not effectively transduced due to a lack of or inadequate expression of the LDLR receptor, through which it gains entry into the cell [111]. The activation of T cells, on the other hand, improves transduction efficiency. In the case of in vivo gene therapy using viral vectors, pre-existing immunity to the virus can lead to a decrease in the number of gene-modified cells which will, in turn, limit the rate of repopulation in vivo. The period between transduction and re-administration of the gene-modified cells is also critical, especially for stem cells that could differentiate, limiting homing back to the target site. To protect the structure of their genomes, eukaryotic cells have defence mechanisms that cause the silencing of alien transposable elements or retroviruses, therefore presenting the possibility of induction of silencing mechanisms that could render the therapeutic gene ineffective due to a lack of expression [112].

## 4. HIV Gene Therapy Delivery via Viral Vectors

The most common viral vector in use for ex vivo gene delivery is the lentiviral vector, due to its ability to transduce both dividing and non-dividing cells. Initially, gammaretroviral vectors were employed for gene delivery; however, their potential to integrate near protooncogenes and cause insertional mutagenesis became evident. Since then, there have been multiple attempts aimed at making viral vectors safer and more effective for gene delivery. The choice of the viral vector is determined by the type of cell to be transduced, viral vector characteristics, the size of the gene of interest (transgene) and the desired effect. Table 1 below gives a summary of the cargo payload (insert size) for different viral vectors.

### 4.1. Gamma Retroviral Vectors (γ RV) in HIV Gene Therapy: Complications and Approaches to Enhance Safety

Gamma retroviral vectors were the among the first viral vectors to be used for effective gene therapy. These vectors transduce actively dividing cells in a cell-cycle dependent manner and therefore do not transduce quiescent cells. Retroviruses were initially the vector of choice because their use resulted in the stable integration of the transgene into the host genome. This benefit, though, was limited by a series of genotoxic events that were observed in approximately 10% of the patients [117], which manifested as various forms of leukemia. γ retroviruses, for example the murine leukemia virus, preferentially integrate near strong enhancer regions [118], transcriptional start sites [119], CpG islands and DNAse-I hypersensitive sites [120] and can cause insertional mutagenesis when they integrate in the proximity of proto-oncogenes. These safety concerns led to the suspension of these clinical trials and, consequently, attempts have been made to improve their safety profile [121]. It is now known that, during the retroviral life cycle, the pre-integration complex (PIC) interacts with various host transcription factors, notably the bromodomain and external terminal family of proteins (BET proteins) [122] and the integrase enzyme, resulting in integration into the host genome. BET proteins function by tethering the viral PIC to host chromatin and, therefore, the development of vectors that are BET-independent can alter their integration profile [123]. Indeed, several groups have demonstrated the efficacy of this approach [117,124,125] in cell lines and murine models. Another approach could be the modification of the integrase enzyme to direct it away from integrating near proto-oncogenes [125]. Attempts at creating self-inactivating RV vectors have been less successful than in lentiviral vectors, since they still retain their capacity for insertional activation of oncogenes [126,127].

### 4.2. Lentiviral Vectors for Stable Gene Expression: Challenges and Strategies to Mitigate Them

Lentiviruses are a type of retrovirus which, in addition to the structural genes *env*, *pol* and *gag*, also have the accessory genes, such as *vpr*, *vpu*, *nef* and *vif*, and the regulatory genes, such as *tat* and *rev*. The best-known lentivirus is the human immunodeficiency virus (HIV).

Engineered lentiviruses are considered suitable for gene transfer because they can transduce both dividing and non-dividing cells in a stable way. This difference from other retroviruses could be due to the fact that the PIC of most retroviruses requires the breakdown of the nuclear membrane (during mitosis), so as to allow access to the nucleus while lentiviral PIC enters the nucleus through active transport via nucleoporins and importins [128]. PIC is an assembly of viral cDNA, some viral proteins from the reverse transcription complex and host cell proteins [129]. Integration of the viral DNA after reverse transcription is mediated by the enzyme integrase and PIC. One study found that PIC preferentially targets regions of open chromatin near the nuclear pore, excluding the internal regions of the nucleus and the peripheral regions of the nuclear lamina. Transcriptionally active genes at the periphery of the nucleus are associated with the nuclear pore complex (NPC) and this influences HIV-1 gene expression [130]. Lentiviruses preferentially integrate with genes actively undergoing transcription and the pattern of integration is supported by the target cell transcriptional program [131]. The host cell protein, lens epithelium-derived growth factor (LEDGF/p75) interacts directly with viral integrase and, without it, integrase fails to move into the nucleus. It is now known that LEDGF/p75 links integrase to chromatin [132].

#### 4.2.1. Generations of Lentiviral Vectors

Lentiviral vectors (LVs) have gone through several modifications since their discovery as vehicles for gene therapy (Figure 4). The first generation of LVs comprised the transgene construct, the envelope construct and a packaging construct with *gag*, *pol* and all regulatory and accessory genes. The change from HIV env to VSV-G env or any other suitable viral envelope, (known as pseudotyping), allows for the efficient transduction of a wide variety of cells, although VSV-G causes the poor transduction of resting [133] lymphocytes and HSCs. The second-generation LVs comprise the transgene construct, the envelop construct expressing VSV-G and a packaging construct with *gag*, *pol* and the regulatory genes *tat* and *rev*. All the accessory genes are removed [134]. The third-generation LVs have a modification in the viral promoter in the transgene construct where the U3 region has been modified by deleting part of the sequence from −418 to −18, leaving only 18 bp (400 bp removed) and therefore creating self-inactivating (SIN) vectors. The 5′LTR therefore has 18 bp U3, R and U5 regions and a PolyA tai. This enables the vector to still be able to carry the transgene cassette while remaining transcriptionally inactive [135]. The deleted section of the U3 is replaced by a heterologous promoter, usually the cytomegalovirus (CMV) promoter or cellular promoters like elongation factor 1a (EF1α). The 3′ LTR, U3 region is also deleted, to prevent reconstitution with the 5′LTR via homologous recombination during the transfection of 293T cells. The vector construct also has the non-coding domains cPPT (central poly purine tract), which improves the efficiency with which RNA is packaged into a capsid, and WPRE (Woodchunk hepatitis post-transcriptional regulatory element), which enhances post-transcriptional processing of the transgene [134]. The packaging construct is split into a *gag*–*pol* construct and a *rev* construct with *tat* being deleted, as its transactivating loop has been removed from the 5′LTR [131,134]. The third-generation systems are therefore assembled from four plasmid constructs, as shown in Figure 4 below. Having three separate packaging constructs reduces the chances of recombination to form replication-competent lentiviruses during plasmid amplification and viral vector production and reduces chances of problems associated with promoter interference [136,137]. Third generation lentiviral constructs have been further redesigned by modifying the plasmid carrying the transgene into what can be considered as fourth generation lentiviral vectors. In this system, called LTR1 or PBS1 (Primer Binding Site1), the 5′LTR has been removed and the RNA signals (PBS-Y-RRE) have been placed downstream of the 3′LTR. These signals are therefore present during vector production but are lost during reverse transcription and are not copied with the transduced transgene, therefore further enhancing their safety [138,139].

#### 4.2.2. Optimizing Lentiviral Transduction

The effective lentiviral transduction of human cells can be limited by restriction factors produced by human cells to restrict lentiviral infection, like tripartite motif protein 5 alpha (TRIM5α), tetherin and apolipoprotein B mRNA-editing enzyme catalytic polypeptide-like 3 (APOBEC3) [140]. One of the factors that has been shown to cause inefficient lentiviral gene delivery to quiescent memory T cells is the restriction factor SAMHD1 (sterile alpha motif and histidine-aspartate (HD) domain-containing protein 1). SAMHD1 is a cellular deoxynucleotide triphosphohydrolase that blocks reverse transcription [141,142]. The *vpx* gene, which is encoded by HIV-2, has been found to counter SAMHD1 restriction ability [143]. Indeed, one study found that *vpx* increased gene therapy delivery in all conditions they tested, but the greatest effect was when the gene therapy targeted the steps before reverse transcription [144].

The optimisation of ex vivo culture conditions is a useful approach for enhancing the rate of gene transfer. The use of reagents like Polybrene [145], Protamine Sulphate [146], Retronectin [147], ecotropic receptor boosters, magnetic beads like Lenti-X accelerator, Vectofusin-1 [148,149], LentiBOOST [150] and Staurosporine [151] have been found to enhance the transduction process. Using foetal bovine serum (FBS) was found not to be satisfactory in the transduction of CD34 cells using lentiviral vectors, because it causes an increase in transduction in progenitor cells but a decrease in HSCs. Additionally, there are concerns regarding immune risks, due to the production of antibodies against its xenogeneic components [152]. Various serum-free media have been developed that are suitable for the transduction of different types of stem cells, for example, StemSpan Serum-Free Expansion Medium (SFEM) from Stem Cell Technologies, Vancouver, BC, Canada. Adding knockout serum replacement (KSR) medium to the StemSpan media has been found to enhance transduction efficiency in primary CD34+ cells and peripheral blood mononuclear cells (PBMCs) [153].

#### 4.2.3. Lentiviral Vector Silencing

Like gamma retroviral vectors, though to a lesser extent, lentiviral vectors are susceptible to transgene silencing and the variegation of transgene efficiency, which is due to the interaction of the transgene with its immediate genomic neighbourhood in the host genome (chromosomal position effects). A number of host and vector characteristics have been postulated to cause this effect, including DNA methylation and histone modifications, including those mediated by the Polycomb group of proteins [154]. Strategies that can insulate the transgene against silencing include the use of stronger enhancer promoters, scaffold matrix attachment regions (S/MARS), which protect the enhancer from DNA methylation, and chromatin domain insulators, which suppress repressive position effects [155].

#### 4.2.4. Strategies to Reduce Insertional Mutagenesis

There have been various attempts at reducing the chances of insertional mutagenesis. One strategy is to direct integration to ‘safe genomic harbour sites’ far away from proto-oncogenes. Schenkwein et al. used I-PpoI, a dimeric 18–20 KDa homing endonuclease from the slime mould *Physarum polycephalum*, to direct integration to a highly conserved site, the 28S ribosomal RNA. This is achieved by recognizing a 15 bp site in this region [156]. Another strategy is the production of non-integrating lentiviral vectors by altering the catalytic triad of the integrase core domain (D64, D116 and E152) [157,158,159]. Although the risk of insertional mutagenesis is eliminated, these vectors can only be used for transient gene expression. Gene expression is also lower than in integrating lentiviral vectors [157]. The modification of LEDGF/p75, whose function is to direct lentiviral integration, can allow integration to safer regions or a more random integration which reduces the chances of integration near protooncogenes [160].

### 4.3. Adeno Virus Associated Vectors (AAVs): Challenges Limiting Their Use

Recombinant adeno-associated virus have been the vectors of choice in clinical applications where a transient expression of the transgene (several months to a few years, depending on the turnover of infected cells) is desired. These do not contain viral DNA but are protein-based nanoparticles that are engineered to traverse the cell membrane and deliver the DNA into the nucleus of the cell [161]. These vectors are highly stable and can withstand physical or chemical challenges that arise during manipulation. The viral genome can also be easily manipulated [162] and has a good safety profile. AAVs could carry transgene cassettes of up to 4.5 kb [113]. The major challenge with AAVs, however, is that they are widespread in nature, and therefore anti-AAV immunity is high with the prevalence of neutralizing antibodies for the 12 known serotypes ranging from 20% to 100% in some populations [163]. The administration of these vectors could also evoke post-treatment humoral responses. Various strategies are being applied to overcome capsid immunity, complement fixation and AAV genome sensing [164,165]. One of the strategies is engineering the capsid through rational design, random mutations and capsid shuffling to prevent capsid neutralization. Another strategy is immunosuppression, i.e., the blocking of classical pathways that are implicated in B-cell activation and therefore inhibit humoral response, though humoral response may be limited in the case of ex vivo gene therapy. To reduce AAV genome sensing, TLR signalling can be prevented by depleting CpG dinucleotides in the vector genome. Adeno-associated vectors can be used to transduce a variety of cells including mesenchymal stromal cells [166], neural cells [167] and several others [161], although they are not suitable for hematopoietic cells [162].

Various groups are studying the use of AAVs for the delivery of broadly neutralizing antibodies into skeletal muscle, to provide a long-lasting effect against HIV, either as prevention or treatment, or both. A recent review article analysed their use in HIV infection, and the current challenges and strategies that can be employed to overcome barriers to their efficacy [168].

### 4.4. Ex Vivo Cell Selection and Expansion

Depending on the cell type transduced, the type of vector, the multiplicity of infection (MOI) and the transduction technique used, the number of cells transduced is generally less than what is expected theoretically. Therefore, the transduction process is not 100 percent effective. There are, however, several ways to maximise the transduction efficiency. In addition to the examples provided in Section 4.2.2, using a high MOI results in higher vector copy number (VCN), although this increases the risk of aberrant splicing events [169]. Purifying the vector particles through chromatography can also increase the transduction efficiency [170]. Following transduction, it is beneficial to select for the cells carrying the transgene and expand these cells to enhance their engraftment in vivo. This can be achieved using cell-surface markers [171] or designing the vector to express an antibiotic resistance gene, which will then allow for the preferential selection of cells with the transgene, upon the application of the appropriate antibiotic [170].

The goal in the expansion of transduced HSC is to increase quantity while retaining ‘stemness’. Traditional media that contain serum albumin and cytokines generally support differentiation to mature lineages leading to a loss of capacity to self-renew. The expansion of HSC is preferentially undertaken with a serum-free medium to limit the possibility of contamination. A recent article by Tajer et al. gives a summary of factors added to media to enhance expansion [172]. Several cytokines and growth factors including SCF, FLt3 ligand, TPO (thrombopoietin), IL-3 and IL-6 have been used, although IL-3 is not included in some media because it stimulates the expansion of progenitors rather than HSCs [104,173]. Other compounds have also been found to be useful. These include prostaglandin E_2_ (PGE_2_), which enhances the engraftment of HSCs by improving their homing and self-renewal capacity [174], and Stem Regenin 1 (SR1), an aryl hydrocarbon receptor antagonist [175], which is used in the expansion of both human and murine CD34+, although it has also been found to favour the multipotent progenitors rather than the long-term repopulating stem cells. This contrasts with UM171, a pyrimidoindole derivative, which supports the expansion of stem cells more than the multipotent progenitor cells [176,177,178]. They could therefore be used in combination to promote the expansion of both lineages. Other studies, using murine models, have shown that signalling pathways that regulate the homing and differentiation of stem cells can be modulated. These include the Notch signalling pathway [179], Homeobox genes [180], Wnt [172], among others [174]. The overexpression of Sall4, a zinc-finger transcription factor, results in a rapid and efficient expansion of HSCs (a 50-fold increase in CD34+ cells) in vitro and in vivo in mice models [181].

## 5. Administration of Gene-Modified Hematopoietic Stem Cells: Enhancing Preferential Engraftment of Gene-Modified Cells

The first phase of hematopoietic reconstitution after the infusion of HSCs (up to 6 months) is due to committed progenitor cells (short-term HSCs), which have a limited capacity for self-renewal [182]. After about 6 months, there is a reduction in CD34+ clones, an exhaustion of the first wave of clonal reconstitution and a gradual take-over of reconstitution by long-term HSCs, which are capable of self-renewal and whose effects last even at 3 years post-gene therapy [183]. Short-term HSCs are, therefore, active in the early phase, while the long-term HSCs enter into quiescence during this phase and are activated upon the exhaustion of the short-term HSCs.

### 5.1. Conditioning Regimen for the Clearance of the Hematopoietic Stem Cell Niche

In autologous gene therapy for hematopoietic disorders, the goal is to replace the defective cells with host cells corrected for the specific genetic disorder. In HIV gene therapy, the goal is to replace cells susceptible to infection with those that are resistant, or which are able to combat the virus. Stem cell niches allow the maintenance of stem cells and regulate their function; HSC niches are perivascular in the bone marrow and spleen [184]. The process of clearing the niche for the uptake of transduced HSCs and the elimination of untransduced cells, also called conditioning, has traditionally been performed using chemotherapy and radiotherapy, which is non-specific to the target organ. Conditioning is conducted to deplete the hematopoietic stem and progenitor cells from the niche, to reduce chances of immune rejection in the case of allogenic transplantation and to treat any malignancy in the case of malignant diseases [185]. Conditioning regimens can be classified into myeloablative conditioning (MAC), non-myeloablative conditioning (NMA) and reduced intensity conditioning (RIC). Myeloablative regimens consist of alkylating agents with or without total body irradiation (TBI) and result in pancytopenia and a wearing-out of haematopoiesis within 1–3 weeks, which does not allow autologous hematologic recovery. Stem cell support is therefore necessary for survival [186]. The MAC regimens are associated with toxicity and, on occasions, death, especially if there is a problem with engraftment. Signs of toxicity include organ failure, mucositis, myelosuppression, secondary malignancy, musculoskeletal disorders and cardiac, pulmonary, reproductive and endocrine abnormalities. The long-term mortality rate of those who have gone through MAC is also higher than the general population [187].

Due to these unwanted effects, lower intensity regimens were developed. A lower intensity means reversible myelotoxicity, when compared to non-reversible myelotoxicity induced by MAC. These regimens are NMA and RIC [186]. Non-myeloablative regimens cause minimal cytopenia and do not require stem cell support. Reduced-intensity conditioning, on the other hand, involves the dose of the MAC regimens being reduced by at least 30% and, although it does not cause ablation of stem cells, nonetheless requires stem cell support for it to be practical in the clinic [186,188]. The details of these conditioning regimens have been reviewed [189]. Although RIC is less toxic, it has been found to be less effective for gene therapy because it results in lower gene marking in peripheral blood cells, due either to less efficient clearance of the niche or inefficient induction of immune tolerance to the transgene [190]. In gene therapy, the dose of radiation or chemotherapeutic agent is determined by the disease condition being treated [191] and the patient characteristics, including age and sex. The European Group for Blood and Marrow Transplantation (EMBT) risk score is a tool that is used to assess the risk of transplantation for a cancer patient but can also be extended to any patient requiring transplantation. It is based on five factors: patient age, disease stage, time interval from diagnosis to transplantation, donor type (in terms of HLA typing) and donor–recipient gender combination [192]. Typically, there is increased mortality when a MAC regimen is used beyond the age of 30, and it is not recommended beyond the age of 50. There is also a higher risk of rejection when the donor and recipient are not of the same sex.

### 5.2. In Vivo Selection for Gene-Corrected Hematopoietic Stem Cells

To improve selection for the cells with the transgene, in vivo chemoselection techniques have been adopted for in vivo studies in animal models. Using drug-resistance genes for the preferential in vivo selection of transduced genes requires that the drug-resistance gene be expressed at high levels in transduced cells and not at all or at low levels in the non-transduced HSCs; the selective drug should deplete the majority of the non-transduced cells, have little non-hematopoietic toxicity and not be genotoxic in non-malignant conditions [193]. Genes that have been used for selection in animal models, especially in mice, include multidrug resistance gene (MDR1), dihydrofolate reductase (DHFR), aldehyde-dehydrogenase (ALDH), cytosine deaminase (CDD) [194], glutathione-S-transferase (GST), methylguanine-DNA-methyltransferase (MGMT) and hypoxanthine guanine phosphoribosyl transferase (HPRT) [193,195].

One study used a mutant of O^6^-methylguanine-DNA-methyltransferase (MGMT) which is a DNA alkylating repair enzyme. The mutant, MGMT^P140K^, showed stable and efficient selection of HSCs in non-human primates [196], although the carcinogenic nature of the alkylating agents used (Carmustine-BCNU, for example) was a disadvantage [197]. Another study used the short hairpin RNA (shRNA) knockdown of hypoxanthine guanine phosphoribosyl transferase (HPRT) to achieve the selection of HSCs when these are exposed to 6-thioguanine (6 TG) in murine models. The use of this method is attractive because it allows the transduced HSCs to self-renew, proliferate and differentiate, and the method is used for clearing the niche and for enabling the preferential engraftment of the cells with the transgene [195,198]. The shRNA-inducing drug resistance is only 48 bp long and the amount of 6 TG used for chemo-selection is low, although 6 TG may induce leukemogenesis. This group used an MOI of one and achieved a lentiviral transduction efficiency of 20–30% after two successive rounds of virus exposure and without pre-stimulating with growth factors so as to preserve the “stemness” [195].

### 5.3. Recent Developments in Niche Clearance

A recent development in improving niche clearance and promoting the engraftment of transduced cells is the use of monoclonal antibodies that are conjugated to a drug that is able to clear the niche. These methods allow for the targeted clearance of HSC niches, therefore reducing the side effects associated with the non-specific nature of the conditioning regimens mentioned above and enabling preferential engraftment of the cells with the transgene. To improve the selection of HSCs for example, initial studies in mice utilized CD45-antibody drug conjugates (CD45-ADCs), but it was soon realized that these were not specific to HSCs and resulted in the depletion of the lymphocytes, since CD45 is also found on the surface of lymphocytes and other white blood cells [199,200]. More specific targeting of HSCs has been made possible using CD117-antibody drug conjugates (CD117-ADC), resulting in greater than 99% depletion of endogenous HSCs in mice models [199,201]. The first group that reported the efficacy of this technique in 2019 conjugated CD117 to saporin using a streptavidin linker and tested its effect on NOD SCID Gamma (NSG) mice and non-human primates [199]. Another group used anti-human CD117 IgG1 monoclonal antibody conjugate, AMG 191, and found it to be effective in non-human primates and is currently undergoing ½se 1/2 clinical trials [202]. Another group conjugated the CD117+ to amanitin (MGTA-117) and found it to be highly effective (it depleted more than 95% of HSPCs) in murine models that had acute myeloid leukemia (AML). CD117 is highly expressed in AML cells, and this regimen was found to decrease the tumour burden in these mice [203].

## 6. Clinical Trials Based on Ex Vivo HIV Gene Therapy

As mentioned earlier, gene therapy efforts for HIV have focused on either making the immune system more apt at fulfilling its functions or have aimed to silence or edit the integrated provirus, rendering its replication incompetent or the cells resistant to infection. There have been several clinical trials that have aimed to examine the utility and safety of HIV gene and cell therapy products (Appendix A).

### 6.1. Cell Therapy-Based HIV Clinical Trials

One of the earliest cell therapy studies created a chimeric T cell receptor: CD4-Zeta gene-modified T cells. Results from this early clinical study (NCT01013415) found that, although there were no between group differences for the viral reservoirs, there was, however, a decrease from the baseline of HIV burden in the group that received the gene-modified cells and a trend towards fewer patients with recurrent viremia [204]. AGT103-T is a product that restores the gag-specific CD4^+^ T cell response in persons with chronic HIV disease. Initial Phase 1 clinical trial results (NCT03215004) showed some promising results. Leukapheresis was performed on patients enrolled in the study. Once the cells had been gene-modified and had passed quality control tests, the patients underwent non-myeloablative conditioning with cyclophosphamide at 1 g/m^2^ a week prior to the infusion of the gene therapy product. AGT103-T modified cells were infused at a dose of 2 to 21 million cells per kg body weight. Gene marking of the cells could be observed even 6 months after the infusion of the gene therapy product. No serious adverse events were recorded [205].

SB-728 is a zinc-finger nuclease (ZFN)-mediated, CCR5-modified, autologous CD4^+^ T cell product. The various versions of this study found no serious adverse events when administered, even after repeat doses. There was also better engraftment when cyclophosphamide was used for conditioning. One of the studies found that, although there was a delay in viral rebound, and improved CD8^+^ T cell responses that persisted for more than 6 months, there was no long term effects on the latent reservoir [206].

Another study, NCT03617198, combined a CAR-T cell therapy with ZFN CCR5 modification in a form of dual therapy that ensured that the CD4^+^ cells are resistant to infection and, at the same time, capable of detecting and clearing HIV-infected cells. Finally, NCT04648046 used a lentiviral vector that encodes for bi-specific anti-gp120 CAR molecules (LVgp120duoCAR-T) to target cells expressing HIV gp-120.

### 6.2. Clinical Trials Based on Gene Delivery via Retroviral Vectors

Earlier studies targeting HIV directly used retroviral vectors for the delivery of anti-HIV genes. Syngeneic T cells (from identical twins) were used in a study where a bacterial gene, NeoR, was delivered using retroviral vectors (NCT00001353). The purpose of this study was to prove that syngeneic cells transduced with a gene can actually persist for weeks to months through the division of mature cells, rather than division of prethymic stem cells [207]. In one study involving twins (NCT00001535), syngeneic CD4+ lymphocytes were obtained from the seronegative twin and transduced with a retroviral vector carrying antisense TAR or antisense *Tat*/*Rev* RNA, transdominant Rev protein or a combination of both. Preclinical studies showed that all the anti-HIV vectors used inhibited HIV, with the transdominant *Rev* protein showing greater inhibition [208]. No results of the clinical trial have been posted. A long-term follow-up of all studies involving twins (Gemini Study—NCT04799483) is currently ongoing. Another study employing retroviral vectors, NCT00002221, obtained peripheral blood CD34^+^ cells from HIV-positive patients with non-Hodgkin’s lymphoma scheduled for autologous bone marrow transplantation and divided these cells into three pools. One pool of cells was transduced with a retroviral vector containing two ribozyme sequences “L-TR/Tat-neo”, a second pool was transduced with a control vector ‘LN’ and the third pool remained unmodified cells. These were then re-infused back into the patient. No results for this study have been posted. Apart from the risk of insertional mutagenesis observed with retroviral vectors, there is also the likelihood that the success of this latter study will be hindered by the low proportion of cells carrying the therapeutic gene.

RevM10 is a dominant-negative mutant of HIV-1 *Rev* gene. The transduction of this gene using retroviral vectors into CD4^+^ T cells resulted in the inhibition of HIV replication and persistence even up to 6 months [209]. Results for studies of RevM10 in hematopoietic stem cells (NCT00003942) have not been posted. Retroviral vectors were also used for the delivery of MazF, a Tat-dependent endoribonuclease gene in a Phase I study to evaluate its safety, tolerability and immunogenicity. MazF is derived from *E. coli* and selectively cleaves ACA sequences of mRNA, which are common in HIV. This study found that the single intravenous infusion of autologous CD4^+^ T cells transduced with MazF caused an increase in CD4^+^ and CD8^+^ cells, and this effect persisted for at least 6 months [210].

Ribozymes are catalytic RNA molecules that can target any RNA sequence with a NUX cleavage site, where N is any nucleotide and X is either A, C or U [211]. In a proof of concept Phase 1 study, Rz2, a ribozyme directed towards a GUA sequence near the initiation codon of the tat gene, was delivered via retroviral vectors into syngeneic peripheral blood mononuclear cells (PBMCs) obtained from a seronegative twin’s gene-modified cells and then infused into the seropositive twin [211]. Four patients were enrolled in this study, which demonstrated the persistence of the gene-marked cells, both in the short-term (24 weeks) and long-term (44 months), with no serious side effects being observed during this period. In a related Phase 2 gene therapy trial of an anti-HIV ribozyme transduced into autologous CD34^+^ cells via retroviral vectors, patients received a *tat*–*vpr*–specific anti-HIV ribozyme (OZ1) or placebo. There were no OZ1-related adverse events. Although there was no statistically significant difference in viral load between the OZ1 and placebo group at the primary end point (around week 48), there were significant differences observed until week 40, and CD4^+^ T cells were higher in the OZ1 group than in placebo until 100 weeks [212,213]. In the long-term follow-up study, NCT01177059, 18 out of 68 who were initially enrolled completed the study, and of these, 7 had serious adverse effects, including neoplasms of various types (e.g., basal cell carcinoma, Hodgkin’s disease, Kaposi’s sarcoma, papillary thyroid cancer, skin cancer), which were most likely due to the retroviral vector.

### 6.3. Lentiviral Vector-Based HIV Studies

In the VRX496 (Lexgenleucel-T) study, autologous T cells were modified using a HIV-based lentiviral vector carrying an anti-HIV antisense gene targeting the HIV envelope. The safety and tolerability study, NCT00295477, found the engraftment half-life in blood to be 5 weeks. Some patients had gene-marked cells even up to 5 years. This study demonstrated that gene-modified cells can exert genetic pressure on HIV [214].

Another study that used lentiviral vectors to deliver anti-HIV genes combined a short hairpin RNA (shRNA) against CCR5, which blocks entry of the virus, a human/rhesus macaque chimeric TRIM5α, a natural occurring molecule that disrupts the uncoating of the viral capsid upon entering the cytoplasm, therefore inhibiting HIV infection, and a transactivation response (TAR) decoy, which mimics viral TAR by binding the viral Tat and sequestering it, therefore preventing it from mediating the efficient transcription of the proviral DNA [215]. No results have been posted yet for this clinical trial.

In one group of studies, patients undergoing autologous transplantation for lymphoma were infused with CD34^+^ hematopoietic stem cells that were modified with a lentiviral construct containing a *tat-rev* shRNA, a TAR decoy and a CCR5 ribozyme (LV Rhiv7-shI-TAR-CCR5RZ). Early results in four patients showed the engraftment of gene-modified cells by day 11 post-infusion and the persistence of the vector even at 24 months, together with selection for gene-modified cells following a viremia [216]. Results for the individual studies have not been posted.

Cal-1 is a dual anti-HIV gene transfer lentiviral construct that delivers a shRNA that silences CCR5 (sh5) and, under a different promoter, the coding sequence for a small peptide inhibitor of viral fusion that prevents fusion of the HIV envelope to the host CD4^+^ cell (C46). Several studies of Cal-1 have been conducted in different locations (Appendix A). In NCT01734850, Cal-1 was administered to 13 patients and the safety of the therapy was assessed. Pre-clinical studies of Cal-1 showed the therapy to be effective and safe [60,217,218]. During the clinical trial, Busulfan conditioning was used to clear the niche to allow the engraftment of gene-modified cells. Severe and life-threatening adverse events were observed in the two treatment arms that underwent Busulfan conditioning, with more severe effects observed with two doses of Busulfan compared to a single dose. The arm that did not undergo conditioning had no severe adverse events. The patients are currently being monitored in a long-term follow-up study (NCT02390297). Toxicity related to conditioning is a common challenge to the successful delivery of gene therapy, and it is therefore worthwhile to develop safer and more effective methods of clearing the niche to allow for successful and stable engraftment.

These clinical studies demonstrate the delicate balance between the adequate clearing of the niche to achieve sufficient engraftment, enabling the production of gene-modified cells and the resulting myelotoxicity due to the conditioning.

It is imperative to solve these challenges in order to help advance pre-clinical investigations of HIV cure approaches, such as studies reported for the block and lock approach using si/shRNA [53,64,219]. This group of pre-clinical mouse investigations have reported the successful silencing of HIV by si/shRNAs, most notably shPromA, which targets the tandem NF-KB transcription factor sites in the virus promoter to induce transcriptional gene silencing of downstream viral gene expression and repressive epigenetic modifications (Figure 2) [66,220]. In combination with the Cal-1 vector containing the CCR5 shRNA, the promoter-targeted si/shRNA, PromA, is currently being investigated for the development of a combination HIV gene therapy targeting both host CCR5 and virus promoter targets.

## 7. HIV Gene Therapy Clinical Trials Based on In Vivo Delivery

Studies employing the in vivo approach in HIV are mainly at the pre-clinical stage. In vivo approaches involve the introduction of the gene therapy product directly to the patient using either viral vectors like adenoviral vectors, adeno-associated virus vectors or non-integrating lentiviral vectors, or through non-viral methods like nanoparticles. In vivo methods may not provide a long-lasting cure, because there is no stable integration of the transgene with the host’s genetic material, and the delivery to specific cells needs to be optimised to reduce the chances of off-target effects. Gene editing through recombinase or nuclease systems [61], or silencing mechanisms using RNA interference or possibly CRISPR-interference, could use the in vivo approach.

Current clinical trials include a study that used leronlimab, an anti-CCR5 humanized IgG4 antibody that competitively inhibits HIV env attachment to CCR5 by binding to the same attachment site as CCR5 (the extracellular loop-2 and N-terminus domains). This agent was administered subcutaneously, and the group is currently working on using a synthetic AAV vector for delivery. Several versions of this study—NCT00642707, NCT02175680, NCT02355184, NCT02483078, NCT02990858, NCT03902522, NCT02859961 and NCT05271370—demonstrated the safety and efficacy of the product, with patients experiencing mostly mild side effects and efficacy being observed at higher doses of 525 mg and 700 mg [221].

### CRISPR-Cas9 Technology for HIV Cure

CRISPR and its CRISPR-associated nuclease 9 (Cas9) system is a gene editing system that induces double strand breaks at genomic regions as mediated by the guide RNA, which are then repaired by the non-homologous end joining pathway. Various review articles describing pre-clinical studies in HIV have recently been published [222,223]. A recent clinical trial using CRISPR-Cas9 for an HIV cure is NCT03164135, which uses this technology to knockout the CCR5 gene in the HSCs of patients undergoing allogenic stem cell transplantation for haematological malignancies. Initial results, in a patient with acute lymphoblastic leukemia, showed remission with full donor chimerism, and donor cells with CCR5 knockout persisted for 19 months without gene editing adverse events. The percentage of lymphocytes that carried the modification, however, was only 5%, and the team is investigating how to make the process more effective [224]. Another recent CRISPR-Cas9-based HIV gene therapy product is EBT-101 (NCT05144386). This gene therapy is administered as a single intravenous infusion and is delivered via an AAV that uses two guide RNAs to target three locations of the integrated provirus. Three patients were recently enrolled in the Phase 1 Clinical Trials, and the initial results presented by the company (Excision Bio Therapeutics, San Francisco, CA, USA) in the European Society for Gene and Cell Therapy annual meeting in October 2023 showed no dose-limiting toxicities or serious adverse events.

CRISPR-Cas 9 can be used to target the HIV-5′LTR, which is the promoter region, therefore preventing the replication or activation of the latent virus; the CCR5 or CXCR4 region, rendering the cells resistant to infection; or restriction factors that promote HIV replication, to block their activity [223]. Whilst no CRISPR-Cas9 gene therapies are currently approved as an HIV cure, the FDA recently approved the first CRISPR-Cas9 cell-based gene therapy treatment for sickle-cell disease, Casgevy, (https://www.fda.gov/news-events/press-announcements/fda-approves-first-gene-therapies-treat-patients-sickle-cell-disease, accessed 27 January 2024), highlighting the innovative advancements in gene therapy and the potential of CRISPR-Cas9 as an HIV cure approach.

## 8. Conclusions

The long-term or permanent expression of anti-HIV genes and the modification of CD4^+^ and CD34^+^ cells to render them resistant to infection or to allow the disruption of the HIV life cycle are important strategies in the quest to achieve a HIV cure. Although the advent of combination ART has vastly improved patient outcomes and the quality of life of people living with HIV, a cure is desirable, especially for patients who have developed resistance to the current regimen. A product that provides at least a functional cure will likely need to multiplex therapeutic agents with different modes of action, to minimize the chances of resistance and be long-acting, or ideally require a single administration. It is therefore essential to harness the power of current advances in biotechnology to enhance gene therapy approaches and render gene therapy products safer, more economical and long-lasting.

## Figures and Tables

**Figure 1 ijms-25-02771-f001:**
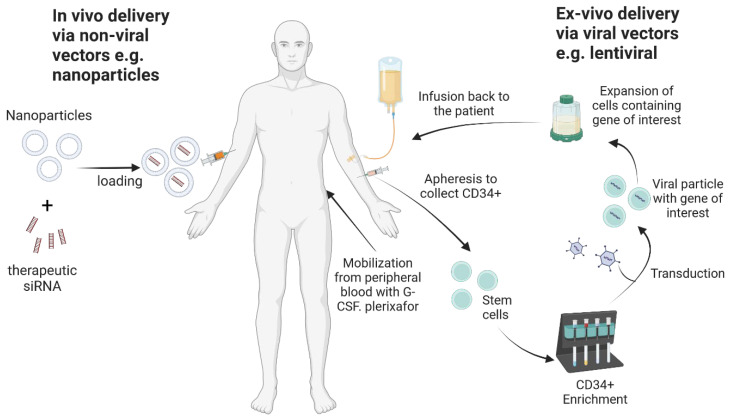
Ex vivo delivery of gene of interest. Created with Biorender.com.

**Figure 2 ijms-25-02771-f002:**
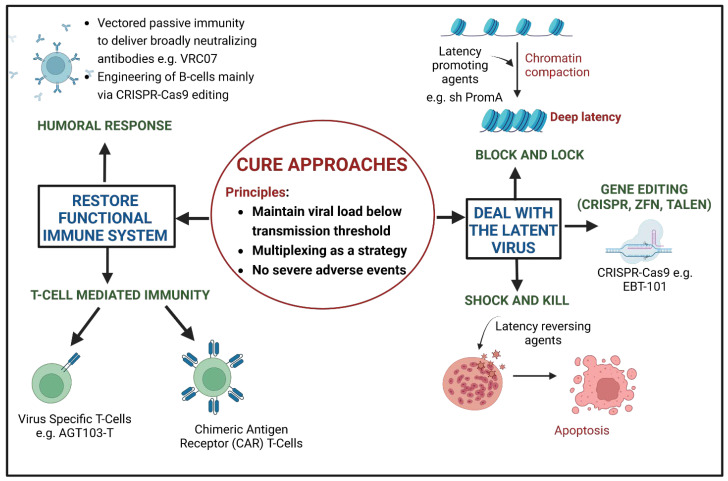
Cure strategies for HIV. CRISPR—Clustered Regularly Interspaced Short Palindromic Repeats; ZFN—Zinc Finger Nuclease; TALEN—Transcription activator-like effector nucleases. Created with Biorender.

**Figure 3 ijms-25-02771-f003:**
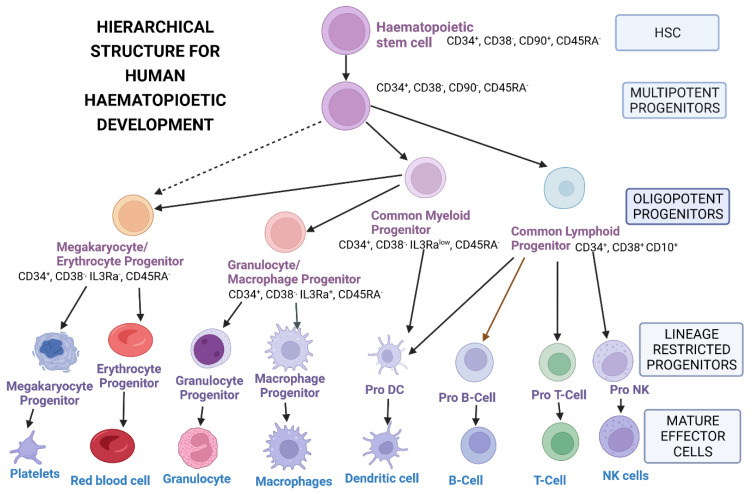
Hierarchical structure for hematopoietic stem cell development. Ref [71]. NK-natural killer cell; DC-dendritic cell. Created with BioRender.com.

**Figure 4 ijms-25-02771-f004:**
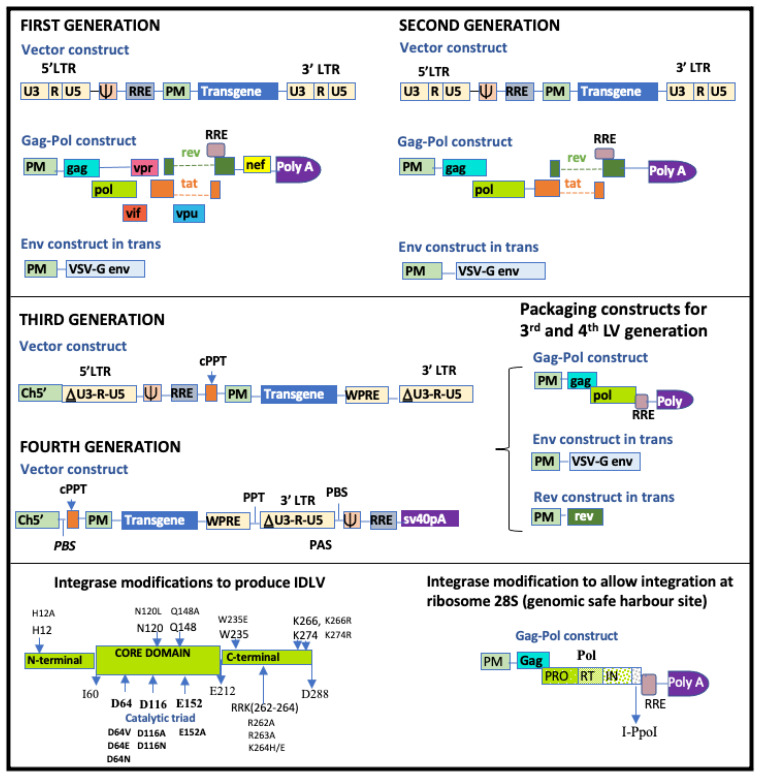
Advances in lentiviral vector generation. The first-generation construct comprises the transgene construct, a pseudotyped envelope (often VSV-G) and a packaging construct with *gag*, *pol* and all regulatory and accessory genes. The second-generation vector are similar to first-generation vectors, except that the accessory genes have been removed. Third-generation vectors are self-inactivating lentiviral vectors with 400 bp deletion in the U3 region. Fourth-generation lentiviral vectors has the 5′LTR removed and the RNA signals (PBS-Y-RRE) placed downstream of the 3′LTR. RRE: Rev responsive element; PM: cell derived or other promoter; cPPT: central poly purine tract; VSV-G: vesicular stomatitis virus type G; WPRE: Woodchunk hepatitis post-transcriptional regulatory element; Y: packaging signal. Ch5′: Chimeric 5′LTR; PAS: Primer Activation Signal.

**Table 1 ijms-25-02771-t001:** Insert size for various viral vectors.

Vector	Type	Insert Size Kilobase (kb)	Reference
Adenoviral	First	4.5	[113]
	Second	10.5	
	Third	36	
Adeno-Virus Associated Vectors(AAVs)		5	[113,114]
Bi-directional vectors	Dual gene cassettes—10	[113]
Self-complementary AAVs (double stranded genome packaged)	Approx. 2.5	[113]
Gamma retroviral		5.5 optimal, but up to 10	[115]
Lentiviral	First	Up to 10	[116]
Second	Up to 10	
Third	Up to 10	
Fourth	Up to 10

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
