# Peer review of "Advances in HIV Gene Therapy"

_ijms, 2024, doi:10.3390/ijms25052771_

Round 1

Reviewer 1 Report

Comments and Suggestions for Authors

This review manuscript aims to emphasize the various stages of ex vivo gene therapy, its current status, and the challenges it faces, particularly in the context of HIV gene therapy. The authors have presented a wealth of information in a straightforward manner. However, there is a lack of in-depth commentary and summaries on related topics. The information regarding recent advances in gene therapy is insufficient. For instance, in the lentivirus section, the authors discuss the third-generation and fourth-generation vectors but do not provide an update on their current applications in gene therapy, which gene therapy drugs were based on the new vectors. The products approved basing on Adeno-associate virus should be further updated.

Furthermore, there is a significant amount of unrelated content in this manuscript. For instance, sentences 351-353 are not pertinent to the main topic. Additionally, there are too many figures (Figure 1, Figure 4, Figure 5, and Figure 6) that may not be necessary for this manuscript.

In summary, the authors have not provided a concise summary and instead seem to have listed a series of common knowledge points. This manuscript requires better organization in terms of both content and writing.

Comments on the Quality of English Language

The authors should try to make the writing more clearly and concisely, e.g. in the abstract section, it may be revised like below “In the late 1990s and early 2000s, gene therapy showed promise for curing heritable diseases, but genotoxic events led to a pause in clinical trials. Recent advances in genetic engineering reignited interest and led to the approval of the first gene therapy for genetic mutations in 2017. Gene therapy (GT) can be administered in either in vivo or ex vivo methods. Using an ex vivo approach in gene therapy offers the advantage of characterizing and selecting gene-modified cells with desired properties before patient administration………………”

Author Response

We thank the reviewer for their comments and have made extensive changes to address the points. Please see the attachment.

Reviewer 2 Report

Comments and Suggestions for Authors

The review structure needs corrections. Some parts seems to be artificially composed

Some remarks:

1.     The manuscript structure needs corrections. If placenta is mentioned as a main source of hematopoietic and MSCs, then neuro stem cells does not fit to this review. The review is mostly about the HSC and MSC. However, other types of the cells are also mentioned (page 13) as well as many disease types (page 16) not only HIV. If HSC and MSC are chosen then the information should be related to the use of these cells.

2.     The numbering of sections and subsections all over the review needs strong corrections, for example, page 2 lines 73-74:

1.     Harvesting cells of interest.

1.1.Hematopoietic cells….

1.2.MSC….

1.3.Source of the cells…

2.     Modification with the…..

Now it is not clear which subsection belongs to which section and the whole review is losing a unified structure.

3.     The section - 4. Re-administration to the Patient - is too long and needs subsections.

4.     It is not clear, why HIV is chosen for the more detailed description in this review. There are many other viral diseases. The review is too long and the HIV part can be separated to another review.

Author Response

(The authors gave the same response as above.)

Round 2

Reviewer 1 Report

Comments and Suggestions for Authors

The authors have addressed the major concerns raised.  

Comments on the Quality of English Language

e.g. Line 51,   if HIV Gene Therapy Cure approaches is revised to Cure approaches for HIV via Gene Therapy, the expression would be clearer. 

Author Response

Please see the attachment below.

Reviewer 2 Report

Comments and Suggestions for Authors

The manuscript has been significantly improved, became more understandable. Some English corrections still required, particularly in titles of sections and subsections. Titles should give all information necessary to understand what the section content is about.

1.       Start from the section 2, it should be “Cure approaches of HIV Gene Therapy”.

2.       3.1 – should be “Sources of ….”

3.       3.1.1 – should be “…marrow-derived…” and all subsections.

4.       3.2 – enrichment of what?

5.       3.3. - modification of what? With only one gene?

6.       6. – gene therapy of what and where? Already used in practice? Or just some possibilities in model systems? No explanations.

The titles such as “Gene therapy” or “Viral vectors” and so on, do not give a clear information of what will be in this section.

7.       4.2.1 –  “Generations of …..

8.       4.3 – the title is bad, should be corrected “ Adeno virus-associated vectors”

9.       5. Administration where and for what?

10.   5.1 – what the authors want to condition and for what purpose? 5.2 - …of….

11.   5.3 – not clear at all what the authors want to say. It is probably a Google translator work…

12.   6. – all titles are not clear and should be corrected. 6. – Clinical trial of …….6.1 –  should be: Cell-based HIV studies” and so on.

13.   7. Should be - Clinical trials of….

14.   The table 2 is extremely large and takes almost half of the article and can be moved to the supplementary part.

Comments on the Quality of English Language

The English language is quite OK, just titles of sections and subsections should be corrected. 

Author Response

Please see the attachment below.
